

# Real-time monitoring of superoxide anion radical generation in response to wounding: electrochemical study

Ankush Prasad[1,2,*], Aditya Kumar[1], Ryo Matsuoka[3], Akemi Takahashi[4], Ryo Fujii[4], Yamato Sugiura[4], Hiroyuki Kikuchi[4], Shigeo Aoyagi[3], Tatsuo Aikawa[5], Takeshi Kondo[5], Makoto Yuasa[5], Pavel Pospíšil[1] and Shigenobu Kasai[2,4,*]

[1] Department of Biophysics, Centre of the Region Haná for Biotechnological and Agricultural Research, Faculty of Science, Palacký University, Olomouc, Czech Republic
[2] Biomedical Engineering Research Center, Tohoku Institute of Technology, Sendai, Japan
[3] Hokuto Denko Corporation, Tokyo, Japan
[4] Graduate Department of Environmental Information Engineering, Tohoku Institute of Technology, Sendai, Japan
[5] Department of Pure and Applied Chemistry, Tokyo University of Science, Noda, Chiba, Japan
[*] These authors contributed equally to this work.

Corresponding authors
Ankush Prasad,
prasad.ankush@gmail.com,
ankush.prasad@upol.cz
Shigenobu Kasai, kasai@tohtech.ac.jp

## ABSTRACT

**Background**. The growth and development of plants is deleteriously affected by various biotic and abiotic stress factors. Wounding in plants is caused by exposure to environmental stress, mechanical stress, and via herbivory. Typically, oxidative burst in response to wounding is associated with the formation of reactive oxygen species, such as the superoxide anion radical ($O_2^{\bullet-}$), hydrogen peroxide ($H_2O_2$) and singlet oxygen; however, few experimental studies have provided direct evidence of their detection in plants. Detection of $O_2^{\bullet-}$ formation in plant tissues have been performed using various techniques including electron paramagnetic resonance spin-trap spectroscopy, epinephrine-adrenochrome acceptor methods, staining with dyes such as tetrazolium dye and nitro blue tetrazolium (NBT); however, kinetic measurements have not been performed. In the current study, we provide evidence of $O_2^{\bullet-}$ generation and its kinetics in the leaves of spinach (*Spinacia oleracea*) subjected to wounding.

**Methods**. Real-time monitoring of $O_2^{\bullet-}$ generation was performed using catalytic amperometry. Changes in oxidation current for $O_2^{\bullet-}$ was monitored using polymeric iron-porphyrin-based modified carbon electrodes ($\varphi = 1$ mm) as working electrode with Ag/AgCl as the reference electrode.

**Result**. The results obtained show continuous generation of $O_2^{\bullet-}$ for minutes after wounding, followed by a decline. The exogenous addition of superoxide dismutase, which is known to dismutate $O_2^{\bullet-}$ to $H_2O_2$, significantly suppressed the oxidation current.

**Conclusion**. Catalytic amperometric measurements were performed using polymeric iron-porphyrin based modified carbon electrode. We claim it to be a useful tool and a direct method for real-time monitoring and precise detection of $O_2^{\bullet-}$ in biological samples, with the potential for wide application in plant research for specific and sensitive detection of $O_2^{\bullet-}$.

## INTRODUCTION

The formation of reactive oxygen species (ROS) in plants is an unavoidable consequence of photosynthesis (*Ledford, Chin & Niyogi, 2007*; *Alessandro et al., 2011*; *Foyer & Shigeoka, 2011*; *Laloi & Havaux, 2015*). The introduction of molecular oxygen into the environment by photosynthetic organisms during the evolution of aerobic life is associated with the formation of ROS (*Tripathy & Oelmüller, 2012*). Plants growing in a fluctuating environment are exposed to various biotic stresses such as bacteria, viruses, fungi, parasites, insects, weeds, etc. and abiotic stresses such as fluctuations in temperature, salinity, water, radiation, toxic chemicals and mechanical stress which are closely linked to higher ROS production. The chloroplasts, mitochondria, and peroxisomes are among the chief organelles involved (*Elstner, 1991*; *Foyer & Harbinson, 1994*; *Asada, 1996*; *Turrens, 2003*; *Liu et al., 2007*; *Murphy, 2009*). As a response, ROS, including the superoxide anion radical ($O_2^{\bullet-}$), hydroperoxyl radical ($HO_2^{\bullet}$), hydrogen peroxide ($H_2O_2$), hydroxyl radical ($HO^{\bullet}$), singlet oxygen ($^1O_2$), peroxyl radical ($ROO^{\bullet}$), hydroperoxide ($ROOH$) and alkoxyl radical ($RO^{\bullet}$), are produced (*Miller, Shulaev & Mittler, 2008*; *Gill & Tuteja, 2010*; *Foyer & Noctor, 2005*; *Asada, 2006*; *Miller et al., 2009*; *Bhattacharjee, 2010*; *Choudhury et al., 2013*).

The production of ROS by an oxidative burst is an imperative element of the wound response in algae, plants, and animals (*McDowell et al., 2015*). As a response to wounding, plants release oligosaccharide cell wall fragments, which play an important role in the signaling cascade that initiates an intense, localized production of ROS (*Legendre et al., 1993*; *John et al., 1997*; *Stennis et al., 1998*). Wounding stimulates the production of $O_2^{\bullet-}$, $H_2O_2$ and nitric oxide (NO), which can directly attack encroaching pathogens at the site of the wound (*Murphy, Asard & Cross, 1998*; *Garces, Durzan & Pedroso, 2001*; *Jih, Chen & Jeng, 2003*). In *Arabidopsis thaliana* leaves measured under ambient light conditions, $O_2^{\bullet-}$ and $H_2O_2$ mainly originate from photosynthetic electron transport, predominantly at the site of wounding (*Morker & Roberts, 2011*). The role of NADPH oxidase in ROS production, however, was not completely ruled out. Therefore, the generation of $O_2^{\bullet-}$ and $H_2O_2$ can be attributed to the collective effect of wounding and light stress. $O_2^{\bullet-}$ generation in the root cells of plants in response to wounding has been studied by electron paramagnetic resonance (EPR) spin-trap spectroscopy and epinephrine-adrenochrome acceptor methods (*Vylegzhaninat et al., 2001*). Tiron (4, 5-dihydroxy-1, 3-benzene-disulfonic acid disodium salt) was used, and the tiron semiquinone EPR spectra showed $O_2^{\bullet-}$ generation. The level of $O_2^{\bullet-}$ production in the roots was measured with epinephrine which in the presence of $O_2^{\bullet-}$ is converted to adrenochrome and can be monitored at 480 nm by a spectrophotometer (*Misra & Fridovich, 1972*; *Barber & Kay, 1996*). In addition to spectroscopy, staining with dye, such as tetrazolium dye and nitro blue tetrazolium (NBT), has been used to detect the production of $O_2^{\bullet-}$ *in situ*, with visualization of $O_2^{\bullet-}$ generation as a purple formazan deposit within leaflet tissues (*Wohlgemuth et al., 2002*).

Although various methods, such as EPR spin-trapping spectroscopy (*Von, Schlosser & Neubacher, 1993*), chemiluminescence (*Anderson et al., 1991*), the reduction of NBT and the reduction of the redox protein cytochrome c (*Doke, 1983b*; *Doke, 1983a*; *Doke, 1985*) have been used to detect and monitor $O_2^{\bullet-}$, each of these methods has inadequate specificity and sensitivity. EPR spin-trapping spectroscopy is one of the most sensitive and specific method for ROS detection; however, kinetic measurements are not possible at the current stage of development. Chemiluminescence, also known as ultra-weak photon emission, has been widely used recently as a non-invasive method to understand the involvement of ROS in oxidative radical reactions (*Prasad & Pospíšil, 2012*; *Prasad & Pospíšil, 2013*; *Pospíšil, Prasad & Rác, 2014*); however, limitations with respect to the specificity for particular ROS involvement exists (*Halliwell & Gutteridge, 1989*).

Integration of metalloporphyrins into electropolymerized polymer electrodes have been developed rigorously over the last years because these materials are effective electrocatalysts for chemical as well as photochemical applications (*Bedioui et al., 1995*). Numerous authors have recently tested the potential use of electropolymerized metalloporphyrins as new electrode materials for chemical and biological sensors (*Deronzier & Moutet, 1996*; *Yim et al., 1993*; *Bedioui, Trevin & Devynck, 1996*). In our current study, we provide an experimental approach for the detection of $O_2^{\bullet-}$ by polymeric iron-porphyrin-based modified carbon electrode based on the reaction mechanism presented in Fig. 1A (*Yuasa & Oyaizu, 2005*). Detection of $O_2^{\bullet-}$ by highly sensitive and selective polymeric iron-porphyrin-based modified carbon electrodes was tested in *in vivo* leaf sample subjected to wounding. The current study introduces the use of catalytic amperometric biosensors for the real-time detection of $O_2^{\bullet-}$.

## MATERIAL AND METHODS

### Spinach leaves
Young spinach (*Spinacia oleracea)* leaves were washed twice with deionized water and were dark adapted for 2 h. For each measurement, a fresh spinach leaf of the approximately same age was chosen. All experiments were performed at room temperature under dark conditions to avoid interference from light sources.

### Material and chemical reagents
The 5-(ethoxycarbonyl)-5-methyl-1-pyrroline N-oxide (EMPO) spin trap and capillary tubes used for EPR measurements were obtained from Alexis Biochemicals (Lausen, Switzerland) and Blaubrand intraMARK (Brand, Germany), respectively. The carbon electrodes ($\varphi = 1$ mm) were purchased from BAS Inc., ALS Co., Ltd. (Tokyo, Japan). Superoxide dismutase (SOD), xanthine oxidase and xanthine (X/XO) were purchased from Wako Pure Chemicals Industries, Ltd. (Osaka, Japan), Sigma-Aldrich chemie Gmbh (Munich, Germany) or Sigma-Aldrich Japan K.K. (Tokyo, Japan).
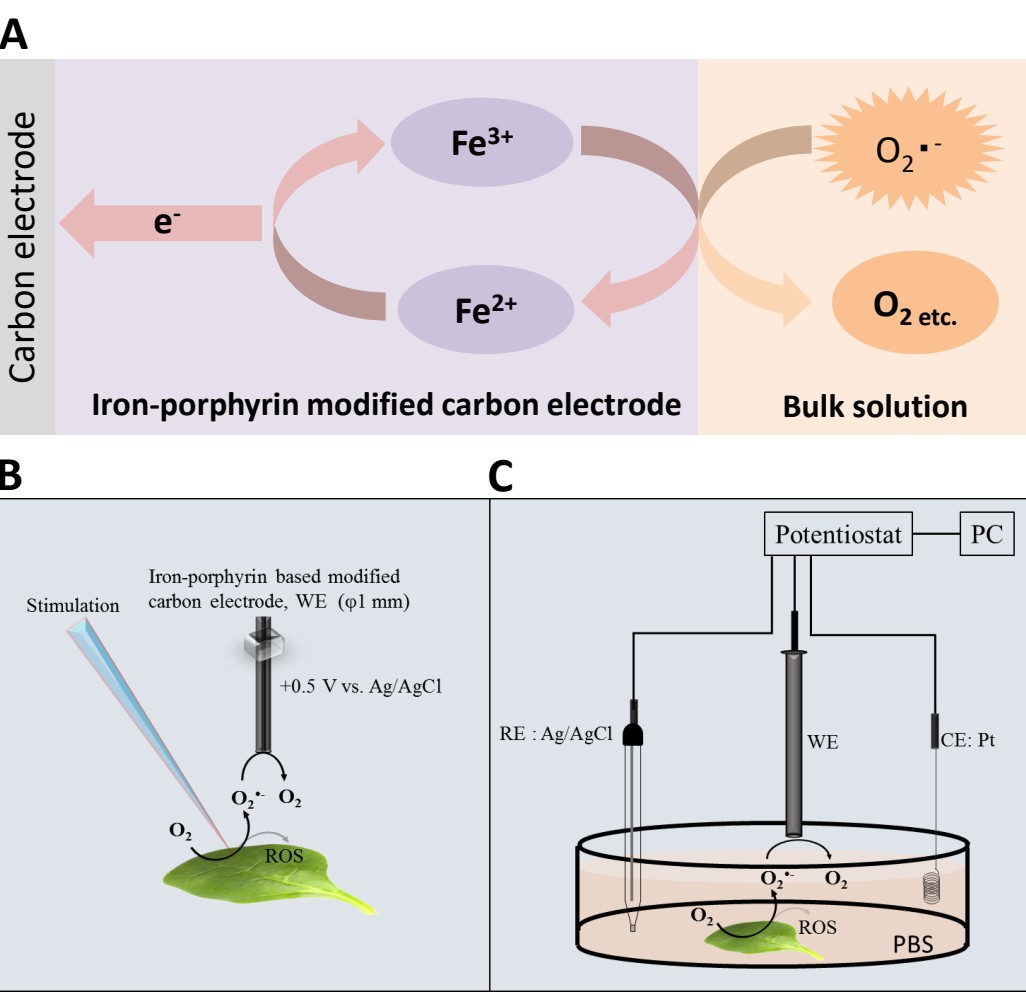

**Figure 1  Reaction mechanism and experimental setup.** (A) Schematic illustration of the reaction mechanism for the amperometric detection of $O_2^{\bullet-}$ using the polymeric iron-porphyrin-based modified carbon electrode depicting the reduction-oxidation cycle leading to generation of the oxidation current. (B and C) Schematic illustration of the experimental setup for the electrochemical measurements. The stimulation was performed using a glass capillary, and the polymeric iron-porphyrin-based modified carbon electron was positioned at a distance of 1 mm using a motor-driven XYZ microscopic stage (B). The *in vivo* generation of $O_2^{\bullet-}$ was measured using a polymeric iron-porphyrin-based modified carbon electron (working electrode, WE), platinum wire (counter electrode, CE) and Ag/AgCl (reference electrode, RE) (C).

## Equipment and methods

Simultaneous measurements of the oxidation current of $O_2^{\bullet-}$ were performed using a potentiostat (HA1010mM4S; Hokuto Denko Co., Ltd., Japan). The polymeric iron-porphyrin-based modified carbon electrodes were positioned 1 mm from the site of injury using a motor-driven XYZ-stage (K101-20MS-M; Suruga Seiki Co., Ltd., Japan) (Fig. 1B). The detection of $O_2^{\bullet-}$ in the X/XO system was performed by EPR spin-trapping spectroscopy using 25 mM EMPO in phosphate buffer.

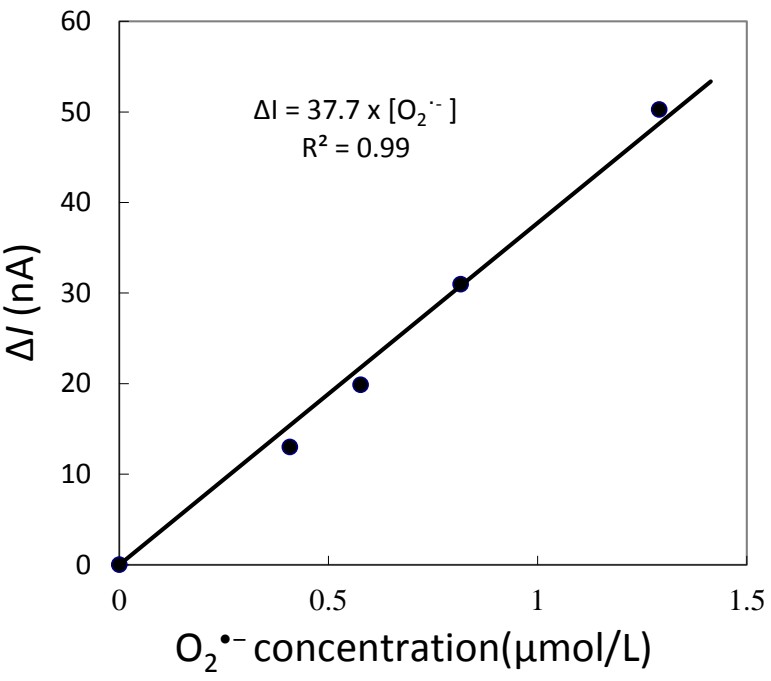

**Figure 2** **Calibration curve.** Changes in oxidation current measured using iron-porphyrin-based modified carbon electrode by exogenous addition of a standard known concentration of $O_2^{\bullet-}$ generated *in situ* using X/XO system in the concentration range of 0.4–1.3 $\mu$M.

## Experimental conditions for real-time monitoring of the oxidation current of $O_2^{\bullet-}$

The electrochemical detection of $O_2^{\bullet-}$ was measured using the X/XO system based on the method described in our recent study (*Matsuoka et al., 2014*) (Fig. 2). The subsequent oxidation current for $O_2^{\bullet-}$ was monitored using polymeric iron-porphyrin-based modified carbon electrodes ($\varphi = 1$ mm) with Ag/AgCl as the reference electrode.

Spinach leaves were fixed on a petri-dish with a diameter of 60 mm using double-sided adhesive tape. A total of 10 mM phosphate buffer saline (pH 7.2) (PBS) was gradually added to maintain a sufficient volume to submerge the whole spinach leaf in PBS. During the measurement, injury was performed using a glass capillary with an inner diameter of about 1.2 mm and wall thickness of 200 $\mu$m as presented in Fig. 1B and Data S1. For data presented in the manuscript, the injury/wounding in spinach leaves were done either one time or multiple times (between 8–10 times) while to visualize the state of leaves, injury/wounding was made one time, five times and 20 times (Data S1). Mechanical injury and mechanical wounding were performed close to the site of the electrode. The oxidation current was measured at +0.5 V vs. Ag/AgCl at room temperature.

## Superoxide anion radical detection using polymeric iron-porphyrin-based modified carbon electrodes

The detection of $O_2^{\bullet-}$ was based on catalytic amperometry using a counter electrode and working electrode. The counter electrode was a platinum wire ($\varphi 0.25 \times 40$ mm), and the working electrode ($\varphi 1$ mm) was a polymeric iron-porphyrin-based modified carbon

electrode. Ag/AgCl was used as a reference electrode. The polymeric iron-porphyrin-based modified carbon electrode acted as an $O_2^{\bullet-}$ detection sensor. The polymeric iron-porphyrin-based modified carbon electrode was prepared by the electropolymerization of 1-methylimidazole-coordinated mesotetra (3-thienyl) porphyrin ($[Fe(im)_2(ttp)]Br$) (*Yuasa & Oyaizu, 2005*; *Yuasa et al., 2005*). Electropolymerization was performed in a two-chamber three-electrode electrochemical cell by potential cycling from 0 to +2.0 V vs. $Ag/Ag^+$ with a potential sweep rate of 50 mV $s^{-1}$. After rinsing with dichloromethane, the polymeric iron-porphyrin-based modified carbon electrode was obtained (*Yuasa & Oyaizu, 2005*; *Yuasa et al., 2005*). For the basic characterization of the polymeric iron-porphyrin-based modified carbon electrode, a differential pulse voltammogram of the electropolymerized $[Fe(im)_2(ttp)]Br$ complex was recorded in an aqueous electrolyte solution containing 10 mM PBS (pH 7.2) using a high-performance potentiostat HZ-7000 (Hokuto Denko Co., Ltd., Japan) (Data S2).

## RESULTS

### Characterization and sensitivity evaluation of an iron-porphyrin-based modified carbon electrode

The characterization of the polymeric iron-porphyrin-based modified carbon electrode was performed using a differential pulse voltammogram (DPV) (Data S2). The polymerized complex was electroactive, with a mean redox potential at −0.25 V for the $Fe^{2+}/Fe^{3+}$ couple.

### Generation of $O_2^{\bullet-}$ in the chemical system and sensitivity evaluation of the polymeric iron-porphyrin-based modified carbon electrode

The xanthine/xanthine oxidase system is used for the formation of $O_2^{\bullet-}$ by the reduction of molecular oxygen (*Olson et al., 1974*; *Richter, 1979*; *Porras, Olson & Palmer, 1981*). To confirm the formation of $O_2^{\bullet-}$ in the chemical system used in the later experimental procedures, we measured the EMPO-OOH adduct EPR signal (Data S3: A). The intensity of the EMPO-OOH adduct EPR signal in the control (xanthine) and chemical system (X/XO) was also measured (Data S3: B). In the absence of xanthine oxidase, no EMPO-OOH adduct EPR signal was observed, whereas in the presence of XO, an EMPO-OOH adduct EPR signal was observed (Data S3).

To determine the sensitivity of the polymeric iron-porphyrin-based modified carbon electrode, the response of the exogenous addition of a standard known concentration of $O_2^{\bullet-}$ generated *in situ* was measured using X/XO system. A linear increase in oxidation current was observed with an increase in $O_2^{\bullet-}$ concentration. The calibration curve ($\Delta i$ vs $O_2^{\bullet-}$) was found to be linear in the concentration range of 0.4 to 1.3 μM (Fig. 2). This indicates that the sensitivity of the electrochemical sensor is in the range of μM concentration, reflecting changes in the oxidation current in the order of tens of nA (Fig. 2).

### Real-time monitoring of $O_2^{\bullet-}$ generation during wounding of spinach leaves

To validate that there is no interference in the measurement caused by the suspension of spinach leaf in PBS, the oxidation current was measured in a non-wounded spinach

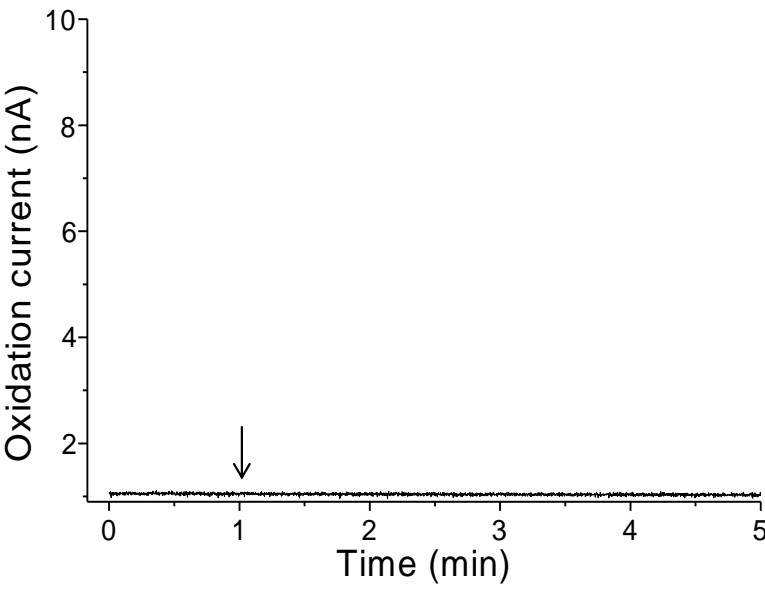

**Figure 3  Real-time monitoring of the oxidation current of $O_2^{\bullet-}$ from spinach leaves.** The kinetics of the production of $O_2^{\bullet-}$ were measured using a polymeric iron-porphyrin-based modified carbon electrode on non-wounded spinach leaves.

leaf suspended in PBS (Fig. 3). No fluctuation in the oxidation current of $O_2^{\bullet-}$ was observed in the non-wounded spinach leaf suspended in PBS, (Fig. 3), whereas a negligible fluctuation was observed with the exogenous addition of SOD (data not shown). These results indicate that these chemical species (PBS and SOD) do not interfere with the measurements. The kinetics of the production of $O_2^{\bullet-}$ were also measured in the chemical system containing no spinach leaves and in the presence of SOD (400 U ml$^{-1}$) indicating no significant fluctuation in oxidation current (Data S4).

Real-time monitoring of the oxidation current for $O_2^{\bullet-}$ was performed in spinach leaves where mechanical injury was stimulated one time using a glass tube (Fig. 4A) and mechanical wounding was done multiple times (Fig. 4B). The wounding in spinach leaves was done one time (4A) and multiple times (between 8–10 times) (4B) using a glass capillary with an inner diameter of about 1.2 mm and wall thickness of 200 μm. The results indicate that the $O_2^{\bullet-}$ production increased considerably with the dose of mechanical injury (Fig. 4). Furthermore, to visualize the extend of damage to leaf during mechanical injury induced by glass capillary, photograph of leaves showing the physiological state have been presented along with kinetics on real-time monitoring of the oxidation current of $O_2^{\bullet-}$ under experimental condition mentioned in dataset presented (Data S1). To determine the concentration of $O_2^{\bullet-}$ generated in mechanically injured spinach leaves, the calibration curve was established for various concentrations obtained using standard X/XO system (Fig. 2). A maximum oxidation current ($\Delta i$) of 1.5 nA (at time span, 60 s) and 7.5 nA (at time span, 300 s) was observed in mechanically injury made at a minimal dose (Fig. 4A) and at multiple sites (Fig. 4B). Based on the data obtained and the maximum

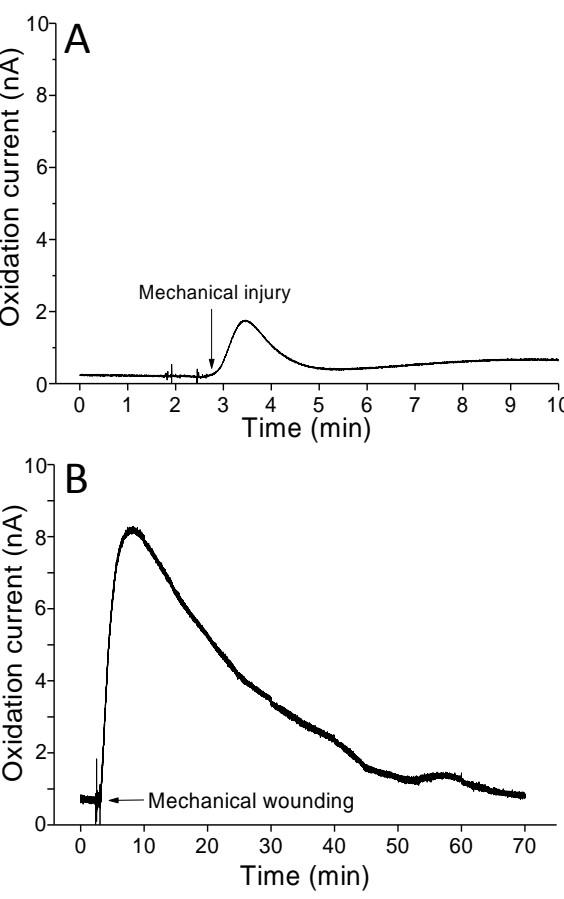

**Figure 4  Real-time monitoring of oxidation current for $O_2^{\bullet-}$ during wounding.** The kinetics of the production of $O_2^{\bullet-}$ were measured using a polymeric iron porphyrin based modified carbon electrode during wounding in spinach leaves. The wounding in spinach leaves was done one time (A) and multiple times (B) close to the site of electrode during the measurement and oxidation current for $O_2^{\bullet-}$ was measured.

**Table 1  Calculation.** Superoxide anion radical ($O_2^{\bullet-}$ concentration calculated using standard calibration curve ($R^2 = 0.9918$) (Fig. 2). The total change in oxidation current was found to be 1.5 nA ($\Delta i$) for minimal dose of injury (Fig. 4A) and 7.5 nA ($\Delta i$) for injury at multiple sites (Fig. 4B). The total $O_2^{\bullet-}$ concentration was found to be equivalent to 40 nM (Fig. 4A) and 200 nM (Fig. 4B) at 60 s and 300 s, respectively.

|  | A | B |
|---|---|---|
| $\Delta i$ (nA) | 1.5 | 7.5 |
| $\Delta t$ (s) | 60 | 300 |
| $O_2^{\bullet-}$ (nM) | **40** | **200** |

oxidation current recorded, the $O_2^{\bullet-}$ was calculated and expected production was found to be about 40 nM (4A) and about 200 nM (4B) (Table 1).

In addition, $O_2^{\bullet-}$ generation was also measured in spinach leaves under the effect of wounding at room temperature in presence of exogenous addition of SOD (Fig. 5). In the absence of wounding, as observed during the first minute of real-time monitoring, no considerable change in the oxidation currents of $O_2^{\bullet-}$ was observed. However, wounding

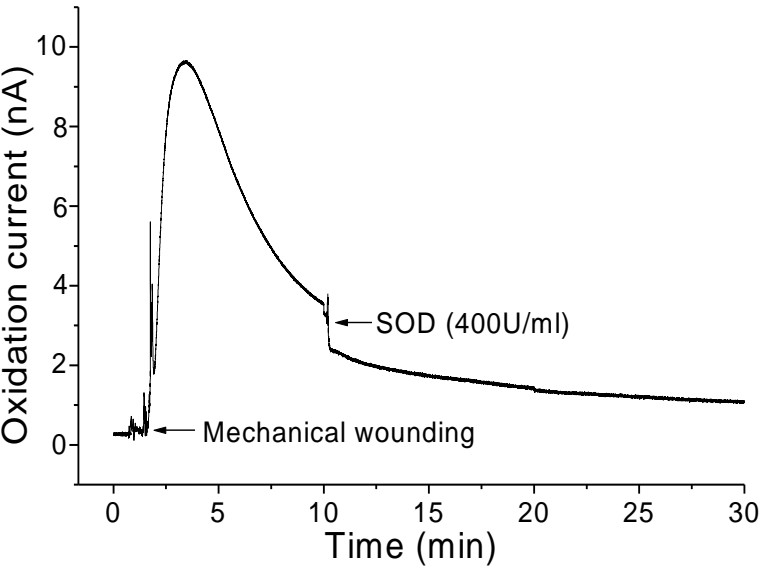

**Figure 5  Real-time monitoring of the oxidation current of $O_2^{\bullet-}$ during wounding.** The kinetics of the production of $O_2^{\bullet-}$ were measured using a polymeric iron-porphyrin-based modified carbon electrode during wounding in spinach leaves. The wounding of spinach leaves was performed during the measurement, and the oxidation current for $O_2^{\bullet-}$ was measured for approximately 30 min. The effect of SOD on the oxidation current was measured in the presence of SOD (400 U ml$^{-1}$) added exogenously during the measurement.

instantaneously resulted in a fast increase in the oxidation current for $O_2^{\bullet-}$ of approximately 10 nA, followed by a gradual decrease, which continued for more than 10 min. To confirm the production of $O_2^{\bullet-}$, the effect of SOD, which leads to the dismutation of $O_2^{\bullet-}$ to $H_2O_2$, on the oxidation current in a wounded spinach leaf was analyzed. The addition of 400 U ml$^{-1}$ SOD suppressed the oxidation current for $O_2^{\bullet-}$ from 3.5 nA to 2 nA (Fig. 5). However, complete suppression of the oxidation current was not observed. The oxidation current, which persisted at approximately 1 nA for a few minutes, can be attributed to rapid $O_2^{\bullet-}$ diffusion to the electrode before its conversion to $H_2O_2$ or to the limited SOD activity at a fixed concentration. The effect of SOD (400 U ml$^{-1}$) added exogenously was also measured at the point of maximum oxidation current where a comparatively higher suppression was recorded (Fig. 6).

## DISCUSSION

In addition to plants, ROS detections have been performed in model system including animals. During recent past, Zuo and coworkers (*2011*, *2013*) presented results on intracellular ROS formation in single isolated frog myofibers during low $P_{O2}$ conditions using dihydrofluorescein (Hfluor), a fluorescein analog of DCFH. Cyt c assay was also used to measure $O_2^{\bullet-}$ in contracting skeletal muscle in pulmonary TNF-$\alpha$ overexpression mice (*Zuo et al., 2014*; *Zuo et al., 2004*). Several mechanisms for the generation of ROS involving $O_2^{\bullet-}$ have been suggested. It has been proposed previously that an NADPH oxidase-like enzyme in the plant plasma membrane is involved in the production of

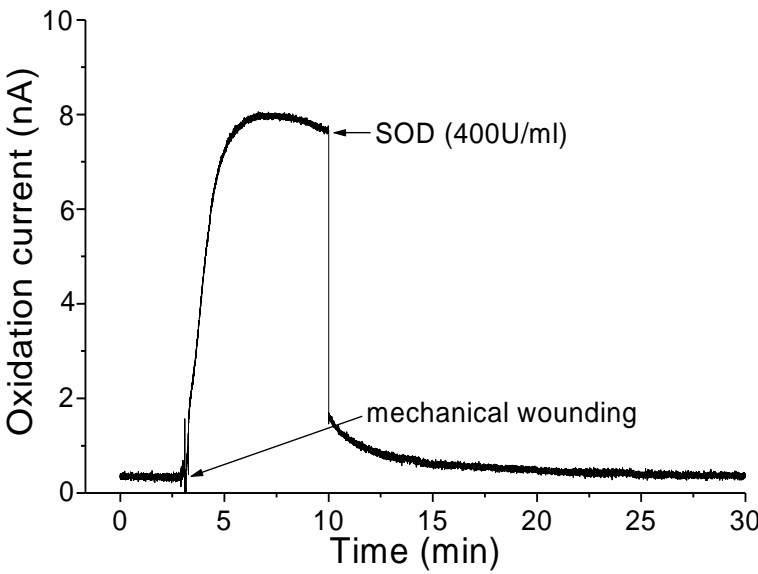

**Figure 6** **Real-time monitoring of oxidation current for $O_2^{\bullet-}$ during wounding.** The kinetics of the production of $O_2^{\bullet-}$ was measured using a polymeric iron porphyrin based modified carbon electrode during wounding in spinach leaves. The wounding in spinach leaves was done during the measurement and oxidation current for $O_2^{\bullet-}$ was measured for a duration of about 30 min. Effect of SOD was measured in the presence of SOD (400 U $ml^{-1}$) added exogenously during the measurement at the point of maximum oxidation current.

$O_2^{\bullet-}$ which is then converted to the more stable $H_2O_2$ during the oxidative burst in response to pathogen attack of plant cells (*Murphy & Auh, 1996*; *Doke et al., 1996*). In skeletal muscle, the major source of ROS especially extracellular $O_2^{\bullet-}$ formation is via the arachidonic acid metabolism through lipoxygenase (LOX) activity (*Zuo et al., 2004*). However, in contrast to the view that wound-induced ROS are primarily produced extracellularly by NADPH oxidase enzymes (*Watanabe & Sakai, 1998*; *Flor-Henry et al., 2004*), it has been recently indicated that wound-induced $O_2^{\bullet-}$ and $H_2O_2$ originate from photosynthetic electron transport measured at wounded sites under ambient light conditions (*Morker & Roberts, 2011*). The $O_2^{\bullet-}$ is produced during the electron transport process by the reduction of molecular oxygen in the chloroplasts and mitochondria. The authors proposed that $O_2^{\bullet-}$ and $H_2O_2$ production is linked to wounding, which is enhanced significantly under light conditions.

A recent report on *Pisum sativum* seedlings proposes a mechanism responsible for oxidative burst during wounding. During mechanical wounding, polyunsaturated fatty acids (PUFA), polyamines, LOX and peroxidases (Prx) are released in the extracellular matrix. Under these circumstances, LOX is involved in the oxidation of PUFA or polyamines, which induces diamine oxidases (DAO) to produce $H_2O_2$, further leading to $O_2^{\bullet-}$ production catalyzed by Prx (*Roach et al., 2015*). A diverse range of organisms use Prx to produce $O_2^{\bullet-}$, e.g., *Triticum sativum* roots (*Minibayeva et al., 2009*), *Castanea sativa* and *Trichilia* seeds (*Roach et al., 2009*; *Whitaker et al., 2010*), liverworts (*Li et al., 2010*), and lichens (*Liers et al., 2011*). Another group of redox enzyme involved in the

wound-induced oxidative burst is DAO (*Rea et al., 2002*; *Cona et al., 2006*; *Yoda, Hiroi & Sano, 2006*; *Angelini et al., 2008*), and cooperation between DAO and Prx is considered important in the wound response. In addition to DAO and Prx, LOX not only generates lipid hydroperoxides (LOOH) but can also generate $O_2^{\bullet-}$ via the oxidation of pyridine nucleotides and therefore considerably contributes to oxidative stress in cells (*Roy et al., 1994*). Lipid oxidation by LOX is an important part of the wound-response because oxylipins, the break-down products of lipid peroxides, can act as effective signaling molecules to rapidly induce transcriptional changes during wounding (*Upchurch, 2008*; *Higdon et al., 2012*). Plasma membrane-bound Prx can utilize PUFA such as linoleic acid released at the wound site for the synthesis of $O_2^{\bullet-}$. Even if LOX is not known to be directly involved in $O_2^{\bullet-}$ production, it was suggested that extracellular LOX may play an important role by competing with Prx for fatty acids and producing reactive electrophiles that coordinate signaling responses (*Doke et al., 1996*).

The electrochemical method with the employment of different modified electrodes have been successfully standardized and applied for detection of varied ROS including $O_2^{\bullet-}$ during the recent past (*Groenendaal Jonas et al., 2000*; *Yuasa & Oyaizu, 2005*; *Yuasa et al., 2005*). The polymeric iron-porphyrin-based modified carbon electrode is a useful tool and provides a direct method for real-time monitoring and precise detection of $O_2^{\bullet-}$ in biological samples *in-situ* (*Yuasa et al., 2005*). The kinetic measurement showing the production of $O_2^{\bullet-}$ for a long range of time (minutes) presented in our study (https://ecs.confex.com/ecs/229/webprogram/Paper70675.html) have been demonstrated and thus opens a new area of investigation which have always been difficult to explore using other available methods such as EPR spin trapping spectroscopy, fluorescence microscopy and other biochemical methods etc. Thus, the electrode has strong potential for wide application in plant research for the specific and sensitive detection of $O_2^{\bullet-}$ and the kinetic behavior in real-time. In addition to points mentioned above, using synthetic porphyrin has an additional advantage. To date, $O_2^{\bullet-}$ sensors based on naturally derived enzyme (e.g., SOD, Cyt. c) have been developed (*Di, Bi & Zhang, 2004*). However, enzymes on the sensor are likely to be denatured. In contrast, the porphyrin based sensor can be used without denaturation. At the same time, the current method is cost effective. Certain limitations exist; the current polymeric iron-porphyrin-based modified carbon electrodes are light-sensitive, which might hinder its photo-electrochemical applicability (*Brett & Brett, 1984*).

## CONCLUSION

In the current study, we present our polymeric iron-porphyrin based modified carbon electrode for application in real-time monitoring and precise detection of $O_2^{\bullet-}$ in biological system. It has strong potential for wide application in plant research for specific and sensitive detection of $O_2^{\bullet-}$.

### Funding

This work was funded by the MEXT-Supported Program for the Strategic Research Foundation at Private Universities, Japan. PP, AP and AK would like to thank the Ministry of Education, Youth and Sports of the Czech Republic (grant no. LO1204 (National Program of Sustainability I) and no. IGA_PrF_2016_013 (Palacký University students project)). The funders had no role in study design, data collection and analysis, decision to publish, or preparation of the manuscript.

### Grant Disclosures

The following grant information was disclosed by the authors:
MEXT-Supported Program.
Ministry of Education, Youth and Sports of the Czech Republic: LO1204, IGA_PrF_2016_013.

### Competing Interests

Ryo Matsuoka and Shigeo Aoyagi are employees of Hokuto Denko Corporation, Tokyo, Japan, and bear no competing interests. In addition, all other authors declare there are no competing interests.

### Author Contributions

- Ankush Prasad conceived and designed the experiments, performed the experiments, analyzed the data, wrote the paper, prepared figures and/or tables.
- Aditya Kumar performed the experiments, wrote the paper.
- Ryo Matsuoka performed the experiments, reviewed drafts of the paper.
- Akemi Takahashi, Ryo Fujii and Yamato Sugiura performed the experiments.
- Hiroyuki Kikuchi, Shigeo Aoyagi, Tatsuo Aikawa, Takeshi Kondo and Makoto Yuasa reviewed drafts of the paper.
- Pavel Pospíšil contributed reagents/materials/analysis tools, reviewed drafts of the paper.
- Shigenobu Kasai conceived and designed the experiments, contributed reagents/materials/analysis tools, reviewed drafts of the paper.

### Data Availability

The raw data has been supplied as a Supplementary File.

### Supplemental Information

Supplemental information for this article can be found online at http://dx.doi.org/10.7717/peerj.3050#supplemental-information.

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
