# Peer review of "Real-time monitoring of superoxide anion radical generation in response to wounding: electrochemical study"

_PeerJ, doi:10.7717/peerj.3050_

## Round 0.1 · original submission · Major Revisions

Please provide a point-to-point detailed response to the reviewers' comments

·

Basic reporting

The authors presented a real-time sensor for ROS detection.
The language is understandable and clear.
It will be advisable to focus on the detection methods in the Intro & background section, and discuss the results and performation of the sensor in the discussion section.
Both of figures 1 and 2 are related little with the theme of the paper and are deletable.

Experimental design

The primary research is within Scope of the journal.
The Research question is well defined, relevant& meaningful.
The investigation design may be improved by addressing some questions.
For example, is the work electrode inserted through the leaf? If so, how depth is it inserted? Will the ROS be derived from the wounding by the electrode itself?
What about the stability as a chemically modified electrode for long-term measurements though authors claimed that it isn’t a problem for measurements lasting only a few hours? It’ll be better that the time-signal curve be provided when the electrode is applied in electrolyte solution.
The sensitivity data should be depicted more detailed in fig.3.
Should the leaf be fluctuated up and down when the wounding is performed? If so, will the depth of the work electrode through the leaf change with the wounding, resulting to the signal fluctuated?
In figure 3, why the curve changed irregularly after the X 50uM was added?

Validity of the findings

As for the results, it’s advisable that the ROS concentrations be measured by the traditional established methods, compared to real time data by the new electrode.
Besides, it should been cautious when the phase “the first time” was used, especially as Ref. 67, 70.

Additional comments

the manuscript should be revised and improved before accepting.

Reviewer 2 ·

Basic reporting

No Comments.

Experimental design

The manuscript could be improved if the measured oxidation current has been converted into O2.- production rate.

The new findings brought by figure 7 as compared with figure 6 was not addressed.

It is unclear how many samples have been examined for figures 3-7.

Validity of the findings

It is not clear how this method is specific for O2.- measurement, which is a major sell point of the work. As shown in figure 6, addition of SOD only partially suppress the oxidation current.

Additional comments

This is an interesting study attemping solve the challenging problem of realtime measurement of ROS production in stressed plants. In particulay, the authors showed that they can record oxidation current, or superoxide anion radical generation, in the spinach leaves subjected to wounding. The study is method-driving and lack of testable hypothesis. In addition, the measurement is highly quantitative without appropriate calibiration, which reduces the significane of the device/method.

Reviewer 3 ·

Basic reporting

The oxidative burst in response to wounding in plants has been shown to associate with the formation of reactive oxygen species (ROS). However, few experimental studies have provided direct evidence of their detection. This study presents a newly developed polymeric iron-porphyrin based modified carbon electrode for the application of real-time monitoring and precise detection of O2•− in biological system in the leaves of spinach (Spinacia oleracea) subjected to wounding. It is claimed to bear a potential for its wide application in plant research for specific and sensitive detection of O2•−.

Experimental design

1. The title of this paper is “Real-time monitoring of superoxide anion radical generation in response to wounding: in vivo electrochemical study”, but the conclusion states that “we present our newly developed polymeric iron-porphyrin based modified carbon electrode for application in real-time monitoring and precise detection of O2•− in biological system both for in vivo and in vitro studies”. Please address this inconsistency.
2. Reference 67 is an in vivo study, thus saying that “The newly developed polymeric iron-porphyrin-based modified carbon electrode is a useful tool and provides a direct method for real-time monitoring and precise detection of O2•− in biological systems for both in vivo and in vitro studies [67]” is inaccurate (page 14, lines 251-254).
3. The Abstract should include 1) numerical data and p values in the result and 2) a brief method description.
4. The authors did a good job in providing a rationale for the study (lines 89-102). The addition of some background information about “polymeric iron-porphyrin-based modified carbon electrode” including its applications and the novelty in the current study will be useful. Also, please focus on discussing the detection method in the Introduction.
5. First paragraph of the Introduction (regarding the background of ROS) can be substantially cut down and combine with the second paragraph to provide a concise view on ROS related to plant wounding. Also, please specify biotic and abiotic factors.
6. How will the presence of electrode itself or the experimental setup (which may consider as environmental stresses) affect ROS production? What correction or data normalization was made to address these confounding factors?
7. The authors only used a plant system to measure real time ROS. The authors should also do experiments using other biology system such as muscle system, or another animal or plant organs to confirm their result. If doing such experiments is not possible sue to the lack of research time and funding, the authors should at least write 5-6 sentences discussing such in other models and following breakthrough on real-time ROS measurement in other models should be cited in references and discussed. These references include: 1) Am J Physiol Regul Integr Comp Physiol. 2013 Jun 1;304(11):R1009-16. doi: 10.1152/ajpregu.00563.2012; 2) Journal of Applied Physiology 111 (3), 898-904, 2011.
8. Please cite Fig. 9 (illustration of the experimental setup) in the Method section.
9. Please see below for the suggested changes in writing and carefully review the manuscript to avoid similar errors.
a. Be consistent when citing the figure. Use either “Fig. 1” or “Figure 1”
b. Please be careful in the use of articles (e.g., the) and commas. For example, “Under homeostatic conditions, the ROS are scavenged…” should be “Under homeostatic conditions, ROS are scavenged…” (page 6, lines 61-62). “…exposure to environmental and mechanical stress and via herbivory” should be “exposure to environmental stress, mechanical stress, and via herbivory” (page 5, lines 28-29).
c. Simply use “photosynthesis” rather than “oxygenic photosynthesis”
d. Change “The oxidative burst in response to wounding is associated with the formation of reactive oxygen species, such as the superoxide anion radical, hydrogen peroxide and singlet oxygen; however, few experimental studies have produced direct evidence of their detection.” to “Typically, oxidative burst in response to wounding is associated with the formation of reactive oxygen species, such as the superoxide anion radical, hydrogen peroxide and singlet oxygen; however, few experimental studies have provided direct evidence of their detection in plants.”
e. Change “For the first time, real-time monitoring of superoxide anion radical generation was performed during wounding by monitoring the changes in the oxidation current using catalytic amperometry utilizing a newly developed polymeric iron-porphyrin-based modified carbon electrode.” to “Real-time monitoring of superoxide anion radical generation was performed during wounding by monitoring changes in oxidation current using catalytic amperometry.”
f. Break this sentence into two: “Catalytic amperometric measurements were performed using newly developed polymeric iron-porphyrin based modified carbon electrode and is claimed to be a useful tool and a direct method for real-time monitoring and precise detection of superoxide anion radical in biological system both for in-vivo and in-vitro studies.”
g. “Arabidopsis” and “in situ” should be italicized.
h. Change “The level of O2•− production in the roots was measured with epinephrine, which in the presence of O2•−, is converted to adrenochrome and can be monitored at 480 nm by a spectrophotometer [32, 33].” to “The level of O2•−production in the roots was measured with epinephrine which, in the presence of O2•−, is converted to adrenochrome and can be monitored at 480 nm by a spectrophotometer [32, 33].”
i. Change “The oxidation current, which persists at approximately 1 nA for a few minutes, can be attributed to rapid O2•− diffusion to the electrode before its conversion to H2O2 or to the limited SOD activity at a fixed concentration.” to “The oxidation current, which persisted at approximately 1 nA for a few minutes, can be attributed to rapid O2•− diffusion to the electrode before its conversion to H2O2 or to the limited SOD activity at a fixed concentration.”
10. Please follow the rule of avoiding “5-words in same sequence” and reword the sentences accordingly to avoid plagiarism. Below is a list of page/line numbers of the plagiarized sentences.
Page 6, lines 50
Page 6, lines 59
Page 6, lines 64
Page 14, lines 245-247

Validity of the findings

No Comments

---

## Round 0.2 · Minor Revisions

Reviewer 3 ·

Basic reporting

no comment

Experimental design

no comment

Validity of the findings

no comment

Additional comments

no comment

---

## Round 0.3 · Minor Revisions

Thanks for the submission. Please address the final comments of the reviewer in a last "minor revision. Thanks.

Reviewer 3 ·

Basic reporting

please see below

Experimental design

please see below

Validity of the findings

please see below

Additional comments

1. Page 10, lines 236-238. The wounding in spinach leaves was done 1 time (4A) and multiple times (between 8-10 tims) using a glass capillary with an inner diameter of about 1.2mm and wall thickness of 200m). The author demonstrated in the supplementary data S1: III. Photographs of Spinach leave showing its state after mechanical injury with glass capillary. Injury was made 1 time (A), 5 times (B) and 20 times (C). I can not find 8-10 times in Material and Methods section. The author should clarify it. In addition, 8-10 tims should be 8-10 times. The parenthesis of 200m) should be deleted.
2. Page 15, lines 355-356. Kinetics of the production of O2•−was measured using polymeric iron porphyrin based modified carbon electrode during wounding in spinach leaves. This
sentence should be “The kinetics of the production of O2•−were measured using a polymeric iron-porphyrin-based modified carbon electrode during wounding in spinach leaves”. Please refer the format of line 352 and line 360.
3. In Figure3, what does the arrow indicate?

---

## Round 0.4 · accepted · Accept

You have addressed all the comments from both reviewers. Congratulations.